# Simultaneous Preference and Metric Learning from Paired Comparisons

**Austin Xu**
Georgia Institute of Technology
Atlanta, GA 30332
axu@gatech.edu

**Mark Davenport**
Georgia Institute of Technology
Atlanta, GA 30332
mdav@gatech.edu

## Abstract

A popular model of preference in the context of recommendation systems is the so-called *ideal point* model. In this model, a user is represented as a vector $u$ together with a collection of items $x_1, \ldots, x_N$ in a common low-dimensional space. The vector $u$ represents the user's "ideal point," or the ideal combination of features that represents a hypothesized most preferred item. The underlying assumption in this model is that a smaller distance between $u$ and an item $x_j$ indicates a stronger preference for $x_j$. In the vast majority of the existing work on learning ideal point models, the underlying distance has been assumed to be Euclidean. However, this eliminates any possibility of interactions between features and a user's underlying preferences. In this paper, we consider the problem of learning an ideal point representation of a user's preferences when the distance metric is an unknown Mahalanobis metric. Specifically, we present a novel approach to estimate the user's ideal point $u$ and the Mahalanobis metric from paired comparisons of the form "item $x_i$ is preferred to item $x_j$." This can be viewed as a special case of a more general metric learning problem where the location of some points are unknown *a priori*. We conduct extensive experiments on synthetic and real-world datasets to exhibit the effectiveness of our algorithm.

## 1   Introduction

Personalized recommendation and ranking algorithms have become increasingly important in recent years, influencing not only the items a user buys and movies he or she watches, but also potentially influencing which job candidates are interviewed, which college applicants are admitted, and even the matching behavior of online dating services. While there are a number of approaches to developing personalized recommendation systems, a particularly common approach uses a classical model for user preference known as the *ideal point model* [1]. In this model a user's preferences are represented as a point $u \in \mathbb{R}^D$ that is embedded in the same space as a set of items $x_1, \ldots, x_N \in \mathbb{R}^D$ (movies, shoes, food, etc). The key model assumption is that the closer an item $x_j$ is to $u$, the more the user will prefer item $x_j$. We note that the ideal point is not necessarily a specific item $x_i$, but rather represents the combination of features that the user most prefers. The ideal point model is an intuitive and interpretable way to model preferences and has been empirically shown to exhibit superior performance compared to other models of preference [2, 3].

In a practical system, the main challenge is to learn the latent $u$ that represents a particular user's preferences. Given a precise quantification of a user's preferences for a number of items, one could infer the distances from $u$ to those items and then easily estimate a good embedding $u$. In practice, however, users find *paired comparison* queries of the form "do you prefer item $x_i$ or item $x_j$" to be far easier to answer [4, 5]. As a result, a number of approaches to learning to rank from such paired comparisons have been proposed in recent years [6–16]. In the specific context of ideal point models, such queries allow the user to reveal which of the two items is closer to their ideal point. There is

now a range of both practical algorithms for estimating $\boldsymbol{u}$ from such queries as well as theoretical treatments analyzing the performance of these algorithms in terms of error bounds and/or sample complexity guarantees [17, 9, 18–26].

While the problem of learning from paired comparisons in the ideal point setting is now well-understood, the vast majority of past work has only examined the case where the user makes judgements under the standard Euclidean distance metric. Assuming a Euclidean metric imposes two main limitations. First, it does not allow for features to interact. In practice, features often complement or compensate for each other. For example, consider the process of purchasing shoes. Each shoe can be described in terms of features such as color, price, materials, etc. An individual may prefer a cost of \$50 and a particular material. However, if the price was set instead to \$200, the user's preferred material may change to reflect the change in price – an effect that cannot be accommodated by a Euclidean (isotropic) metric. Second, the Euclidean metric assumes that all features are of equal importance to the user, which is often not the case. In the shoe purchasing example, a price conscious consumer may prioritize finding the best "bang for their buck," in which case a lower price and higher quality of material would be prioritized over aesthetic features such as color.

To overcome these limitations, we consider the case where the user makes comparisons between items under a *Mahalanobis distance*. Specifically, let $\boldsymbol{M} \in \mathbb{R}^{D \times D}$ be a symmetric positive definite matrix and set $\|\boldsymbol{x}\|_{\boldsymbol{M}} = \sqrt{\boldsymbol{x}^T \boldsymbol{M} \boldsymbol{x}}$. Then $\|\boldsymbol{x} - \boldsymbol{y}\|_{\boldsymbol{M}}$ defines a Mahalanobis distance between $\boldsymbol{x}$ and $\boldsymbol{y}$. This metric captures both feature interactions and the relative significance of those feature interactions via the eigenvalue decomposition $\boldsymbol{M} = \boldsymbol{V} \boldsymbol{\Lambda} \boldsymbol{V}^T$. The eigenvectors specify how features can interact to jointly affect preferences, and the eigenvalues allow for different combinations of features to play a larger or smaller role. See Section 4.2 below for a concrete example.

While a Mahalanobis metric allows for more flexible and powerful models of preference, the appropriate choice of $\boldsymbol{M}$ will in general be unknown *a priori*. In this paper, we develop a novel method to jointly learn both the ideal point and Mahalanobis metric from paired comparisons, which to the best of our knowledge represents the first approach for solving these problems simultaneously. By leveraging the structure of paired comparisons, we develop a simple convex optimization program that estimates $\boldsymbol{M}$ and can then directly solve for $\boldsymbol{u}$. In the process, we also effectively learn the user's ranking of the items. We also explore the possible benefits of a more sophisticated alternating scheme that iteratively refines the estimates of $\boldsymbol{M}$ and $\boldsymbol{u}$. We demonstrate the effectiveness of our approach through experiments on both synthetic and real-world datasets.

## 2  Related Work

Our work naturally builds on the existing literature on learning from paired comparisons, taking particular inspiration from the convex optimization approach to non-metric multidimensional scaling of [17] and the approaches in [9, 18, 21, 26] to developing algorithms for the ideal point setting. We also build on the extensive prior work on *metric learning*. Learning a metric from paired comparisons was introduced in [27], where the authors assume the distance is parametrized by a known matrix $\boldsymbol{A}$ and a weighting matrix $\boldsymbol{W}$ with non-negative diagonal entries. $\boldsymbol{W}$ is learned via a convex program by manipulating the form of diagonal matrix multiplication. Setting $\boldsymbol{A}$ to $\boldsymbol{I}$ does not allow for feature interactions, whereas picking more complex $\boldsymbol{A}$ without overfitting the training data is non-trivial. Similarly in [28], the authors minimize the squared-hinge loss of differences in distances of pairs of items. However, the user is presented with two pairs of items must pick which pair of items is more similar, which is a more complex querying scheme for the user to answer. The same query type is used to learn a metric for images in [29]. The problem of learning low-rank/sparse metrics where all points are known was studied in [30], which established bounds for sample complexity, generalization error, and model identification. The performance of nearest-neighbor-based classifiers have also benefited from learning a Mahalanobis metric that enhances class separation [31], where here class membership is used to inform the learning process.

Metric learning has also been explored in prior work on recommendation systems. For example, [32], [33], and [34] all learn Mahalanobis metrics for ranking given a known reference point and sets of similar and dissimilar items. Using sets of positive and negative items for each user, [35] learns a personalized projection operator for each user and estimates user preference in the learned latent space. In a similar setting, [36] learns transformed ideal points and items directly before learning a metric. Finally, [37] assumes each user has an ideal feature vector where user preference is measured by the inner product of this ideal feature vector with an item's feature vector and develops a feature

selection scheme to account for intransitivity in noisy comparison outcomes while learning an ideal feature vector.

Our work differs from the above in that it uniquely assumes that *both* the metric and ideal point are unknown. Thus, it can be viewed as a more generalized metric learning problem where some of the data are missing. Most existing metric learning papers avoid the problem of knowing a user's preference by assuming a known reference point or utilizing more difficult queries (asking the user to compare two pairs of items). Based on this prior work, it might be unclear if simultaneous recovery of an unknown metric and ideal point is even feasible, but we show that it is indeed possible.

## 3 Observation Model and Estimation Strategy

### 3.1 Observation model

For simplicity, we will begin by considering the noiseless observation model where the user always prefers the item closest to the user's ideal point $\boldsymbol{u}$ under the Mahalanobis distance metric induced by $\boldsymbol{M}$, where $\boldsymbol{M}$ is a symmetric positive definite matrix. For a discussion on how to accommodate noisy paired comparison outcomes, please see the supplementary material. To be more concrete, we let $\boldsymbol{d} \in \mathbb{R}^N$ be the vector with entries $d_i = \|\boldsymbol{x}_i - \boldsymbol{u}\|_{\boldsymbol{M}}^2$. We let $\boldsymbol{y} \in \mathbb{R}^P$ denote our observations, where the $k^{\text{th}}$ element of $\boldsymbol{y}$ denotes the outcome of the $k^{\text{th}}$ comparison (between items $\boldsymbol{x}_{i_k}$ and $\boldsymbol{x}_{j_k}$) and is given by

$$y_k = \text{sign}(d_{i_k} - d_{j_k}). \tag{1}$$

For now we assume that the set of indices $\Omega = \{(i_1, j_1), \ldots, (i_P, j_P)\}$ corresponding to the items compared contains each pair of indices at most once, although our methods could easily be adapted to the case where $\Omega$ is a multiset. We will assume throughout our treatment that the embedding of the items $\boldsymbol{x}_1, \ldots, \boldsymbol{x}_N$ is fixed and known, as in a mature recommendation system. This embedding may correspond to known and interpretable features or be learned from other side information (or even paired comparisons, following a strategy along the lines of [38]).

Before we describe our estimation strategy, a few observations are in order. First, note that $\boldsymbol{y}$ consists of (1-bit) quantized samples of the $N \times N$ matrix $\boldsymbol{\Delta} = \boldsymbol{d}\mathbf{1}_N^T - \mathbf{1}_N\boldsymbol{d}^T$, where $\mathbf{1}_N \in \mathbb{R}^N$ is the vector of all ones. It will also be useful to work with the vectorized version of $\boldsymbol{\Delta}$, which we will denote by $\boldsymbol{\delta}$ and can be written as[1]

$$\boldsymbol{\delta} = (\mathbf{1}_N \otimes \boldsymbol{I}_N - \boldsymbol{I}_N \otimes \mathbf{1}_N)\boldsymbol{d},$$

where $\boldsymbol{I}_N$ denotes the $N \times N$ identity matrix and $\otimes$ denotes the Kronecker product. For conciseness, we will let $\boldsymbol{Q} = \mathbf{1}_N \otimes \boldsymbol{I}_N - \boldsymbol{I}_N \otimes \mathbf{1}_N$.

To index into $\boldsymbol{\delta}$, we map every $(i_k, j_k) \in \Omega$ to a linear index between 1 and $N^2$ defined as $\Gamma = \{(i_k - 1)N + j_k : (i_k, j_k) \in \Omega\}$. We can equivalently write our observation model in (1) as

$$\boldsymbol{y} = \text{sign}(\boldsymbol{\delta}_\Gamma) = \text{sign}(\boldsymbol{Q}_\Gamma\boldsymbol{d}), \tag{2}$$

where the notation $\boldsymbol{\delta}_\Gamma$ and $\boldsymbol{Q}_\Gamma$ indicates the vector or matrix obtained by selecting only the indices/rows indexed by $\Gamma$.

### 3.2 Estimation from unquantized observations

To gain some insight into this problem, we will temporarily ignore the quantization and suppose that we have direct access to $\boldsymbol{\delta}_\Gamma$ – in this case, how might we go about estimating $\boldsymbol{M}$ and $\boldsymbol{u}$?

Consider $d_{i_k} = \|\boldsymbol{x}_{i_k} - \boldsymbol{u}\|_{\boldsymbol{M}}^2$ and $d_{j_k} = \|\boldsymbol{x}_{j_k} - \boldsymbol{u}\|_{\boldsymbol{M}}^2$ for any $(i_k, j_k) \in \Omega$. Then, for the linear index $p$ corresponding to $(i_k, j_k)$, $\boldsymbol{\delta}_p = \|\boldsymbol{x}_{i_k} - \boldsymbol{u}\|_{\boldsymbol{M}}^2 - \|\boldsymbol{x}_{j_k} - \boldsymbol{u}\|_{\boldsymbol{M}}^2$. Observe that when we expand these terms we can cancel the coupled term $\boldsymbol{u}^T\boldsymbol{M}\boldsymbol{u}$, greatly simplifying our subsequent analysis:

$$\begin{aligned} \boldsymbol{\delta}_p &= \|\boldsymbol{x}_{i_k} - \boldsymbol{u}\|_{\boldsymbol{M}}^2 - \|\boldsymbol{x}_{j_k} - \boldsymbol{u}\|_{\boldsymbol{M}}^2 \\ &= \boldsymbol{x}_{i_k}^T\boldsymbol{M}\boldsymbol{x}_{i_k} - 2\boldsymbol{x}_{i_k}^T\boldsymbol{M}\boldsymbol{u} + \boldsymbol{u}^T\boldsymbol{M}\boldsymbol{u} - (\boldsymbol{x}_{j_k}^T\boldsymbol{M}\boldsymbol{x}_{j_k} - 2\boldsymbol{x}_{j_k}^T\boldsymbol{M}\boldsymbol{u} + \boldsymbol{u}^T\boldsymbol{M}\boldsymbol{u}) \\ &= \boldsymbol{x}_{i_k}^T\boldsymbol{M}\boldsymbol{x}_{i_k} - \boldsymbol{x}_{j_k}^T\boldsymbol{M}\boldsymbol{x}_{j_k} - 2(\boldsymbol{x}_{i_k} - \boldsymbol{x}_{j_k})^T\boldsymbol{M}\boldsymbol{u} \end{aligned} \tag{3}$$

If we define

$$\boldsymbol{R} = \begin{bmatrix} — & (\boldsymbol{x}_{i_1} - \boldsymbol{x}_{j_1})^T & — \\ — & (\boldsymbol{x}_{i_2} - \boldsymbol{x}_{j_2})^T & — \\ & \vdots & \\ — & (\boldsymbol{x}_{i_P} - \boldsymbol{x}_{j_P})^T & — \end{bmatrix} \qquad \boldsymbol{S} = \begin{bmatrix} — & (\boldsymbol{x}_{i_1} + \boldsymbol{x}_{j_1})^T & — \\ — & (\boldsymbol{x}_{i_2} + \boldsymbol{x}_{j_2})^T & — \\ & \vdots & \\ — & (\boldsymbol{x}_{i_P} + \boldsymbol{x}_{j_P})^T & — \end{bmatrix}$$

then we can write (3) more concisely as

$$\boldsymbol{\delta}_\Gamma = \mathrm{diag}(\boldsymbol{SMR}^T) - 2\boldsymbol{RMu}, \tag{4}$$

where $\mathrm{diag}(\boldsymbol{A})$ returns the diagonal of $\boldsymbol{A} \in \mathbb{R}^{N \times N}$ as a column vector. For brevity, let $\boldsymbol{a}_M = \mathrm{diag}(\boldsymbol{SMR}^T)$. We now observe that if we observed $\boldsymbol{\delta}_\Gamma$ directly *and* already knew $\boldsymbol{M}$, then we could estimate $\boldsymbol{u}$ by solving a standard least squares problem, resulting in the estimate

$$\widehat{\boldsymbol{u}} = \frac{1}{2}\boldsymbol{M}^\dagger \boldsymbol{R}^\dagger(\boldsymbol{a}_M - \boldsymbol{\delta}_\Gamma). \tag{5}$$

Plugging this estimate into (4), we obtain a simple system of equations that is linear in $\boldsymbol{M}$:

$$\begin{aligned} \boldsymbol{\delta}_\Gamma &= \boldsymbol{a}_M - 2\boldsymbol{RM}(\frac{1}{2}\boldsymbol{M}^\dagger \boldsymbol{R}^\dagger)(\boldsymbol{a}_M - \boldsymbol{\delta}_\Gamma) \\ &= \boldsymbol{a}_M - \boldsymbol{RMM}^\dagger \boldsymbol{R}^\dagger(\boldsymbol{a}_M - \boldsymbol{\delta}_\Gamma) \\ &= \boldsymbol{a}_M - \boldsymbol{RR}^\dagger(\boldsymbol{a}_M - \boldsymbol{\delta}_\Gamma), \end{aligned}$$

where the last equality follows from the fact that $\boldsymbol{M}$ is assumed to be positive definite (and hence full-rank). Rearranging terms, we obtain the more convenient expression

$$\boldsymbol{0} = (\boldsymbol{I} - \boldsymbol{RR}^\dagger)(\boldsymbol{a}_M - \boldsymbol{\delta}_\Gamma). \tag{6}$$

## 3.3 Single-step estimation from quantized observations

Given direct observations of $\boldsymbol{\delta}_\Gamma$, we could immediately estimate $\boldsymbol{M}$ using (6). However, our observations as in (2) are (1-bit) quantized. In this case, using (6), we can instead formulate a constrained optimization problem to jointly estimate an $\boldsymbol{M}$ and a set of distances $\boldsymbol{d}$ (and hence $\boldsymbol{\delta}_\Gamma$) that are consistent with both our observations $\boldsymbol{y}$ and (6). Specifically, we will aim to find a solution that satisfies (6) while minimizing $\ell(\boldsymbol{d})$, where $\ell(\boldsymbol{d})$ is a loss function that encourages $\boldsymbol{d}$ to be such that that the signs of entries of $\boldsymbol{\delta}_\Gamma = \boldsymbol{Q}_\Gamma \widehat{\boldsymbol{d}}$ are consistent with the observed comparisons. For example, one could set $\ell(\boldsymbol{d})$ to be the *hinge loss*:

$$\ell(\boldsymbol{d}) = \sum_{k=1}^{P} \max(0, 1 - y_k (\boldsymbol{Q}_\Gamma \boldsymbol{d})_k), \tag{7}$$

where $(\boldsymbol{Q}_\Gamma \boldsymbol{d})_k$ denotes the $k^{\text{th}}$ element in the vector $\boldsymbol{Q}_\Gamma \boldsymbol{d}$. In the remainder of this paper and in our experiments, we use the hinge loss, but our framework could easily be extended to accommodate any convex loss. Finally, in our proposed approach we also introduce slack variables $\boldsymbol{\zeta} \in \mathbb{R}^P$ to loosen the constraint (6) to improve stability and robustness to noise, and also introduce terms to the objective function to allow for a small amount of regularization on both $\boldsymbol{M}$ and $\boldsymbol{d}$:

$$(\widehat{\boldsymbol{M}}, \widehat{\boldsymbol{d}}, \widehat{\boldsymbol{\zeta}}) = \underset{\boldsymbol{M},\boldsymbol{d},\boldsymbol{\zeta}}{\arg\min}\ \ell(\boldsymbol{d}) + \gamma_1 \|\boldsymbol{\zeta}\|_1 + \gamma_2 \|\boldsymbol{M}\|_F^2 + \gamma_3 \|\boldsymbol{d}\|_2^2 \tag{8}$$

$$\text{s.t.} \quad -\boldsymbol{\zeta} \le (\boldsymbol{I} - \boldsymbol{RR}^\dagger)(\mathrm{diag}(\boldsymbol{SMR}^T) - \boldsymbol{Q}_\Gamma \boldsymbol{d}) \le \boldsymbol{\zeta}$$

$$\boldsymbol{\zeta} \ge \boldsymbol{0}, \quad \boldsymbol{M} \succeq \boldsymbol{0}.$$

The first two constraints aim to enforce (6), while the final constraint enforces that $\widehat{\boldsymbol{M}}$ is symmetric positive semi-definite. The constants $\gamma_1, \gamma_2, \gamma_3$ are parameters set by the user to control the amount of regularization. The above formulation is a convex (semi-definite) program and can be solved by standard tools such as CVX [39, 40].

With $\widehat{\boldsymbol{M}}$ in hand, we can then immediately solve for $\widehat{\boldsymbol{u}}$ via (5). However, since we do not expect our estimate $\boldsymbol{M}$ to be perfect, we instead use the regularized estimate:

$$\widehat{\boldsymbol{u}} = \frac{1}{2}(\widehat{\boldsymbol{M}}\boldsymbol{R}^T \boldsymbol{R}\widehat{\boldsymbol{M}} + \alpha\boldsymbol{I})^{-1}\widehat{\boldsymbol{M}}\boldsymbol{R}^T(\boldsymbol{a}_{\widehat{M}} - \boldsymbol{Q}_\Gamma \widehat{\boldsymbol{d}}), \tag{9}$$

where $\alpha$ is a regularization parameter set by the user. We will see in Section 4 that this simple single-step estimation procedure of estimating $\boldsymbol{M}$ followed by $\boldsymbol{u}$ is surprisingly effective.

### 3.4 Noise considerations

A common noise setting is when paired comparison outcomes are made with respect to differences in distances corrupted by additive iid noise. Such noise may arise from an imperfectly learned embedding or as a way of modeling response errors. While we pose the problem in a noiseless environment for simplicity, the estimation strategy outlined above can be adapted to accommodate such noise by replacing the loss function in (7) with the negative log-likelihood of observing comparison outcomes given a noise model, provided that the log-likelihood function is concave. For example, suppose we assume paired comparison outcomes follow the *Bradley-Terry model* [41], i.e.,

$$\mathbb{P}(\boldsymbol{x}_i \succ \boldsymbol{x}_j) = \frac{e^{-\|\boldsymbol{x}_i - \boldsymbol{u}\|_M^2}}{e^{-\|\boldsymbol{x}_i - \boldsymbol{u}\|_M^2} + e^{-\|\boldsymbol{x}_j - \boldsymbol{u}\|_M^2}} \tag{10}$$

where $\boldsymbol{x}_i \succ \boldsymbol{x}_j$ denotes that item $\boldsymbol{x}_i$ is preferred to item $\boldsymbol{x}_j$ and $y_k$ is the outcome of the $k^{th}$ comparison (with $y_k = -1$ if $\boldsymbol{x}_i \succ \boldsymbol{x}_j$ and $y_k = +1$ if $\boldsymbol{x}_j \succ \boldsymbol{x}_i$). Then, we can replace the loss function in (7) with the negative log-likelihood of observing the $P$ paired comparison outcomes:

$$\ell(\boldsymbol{d}) = \sum_{k=1}^{P} \log(1 + e^{-y_k (\boldsymbol{Q}_\Gamma \boldsymbol{d})_k}), \tag{11}$$

where $(\boldsymbol{Q}_\Gamma \boldsymbol{d})_k$ denotes the $k^{th}$ entry of the vector $\boldsymbol{Q}_\Gamma \boldsymbol{d}$

### 3.5 Alternating estimation

While the single-step approach described above is appealing due to its simplicity, the process of dividing the problem into first estimating $\boldsymbol{M}$ and then estimating $\boldsymbol{u}$ suggests a natural extension of then taking our estimate of $\boldsymbol{u}$ and refining our estimate of $\boldsymbol{M}$, and then iteratively alternating between these two problems to (potentially) improve our estimates. Specifically, after obtaining $\widehat{\boldsymbol{M}}$ and $\widehat{\boldsymbol{u}}$ using (8) and (9), we can set $\widehat{\boldsymbol{M}}^{(0)} = \widehat{\boldsymbol{M}}$ and $\widehat{\boldsymbol{u}}^{(0)} = \widehat{\boldsymbol{u}}$. Then, for iteration $k > 0$, we re-estimate $\boldsymbol{M}$ by solving the following optimization problem that replaces the constraint from (6) with the constraint from (4) to allow us to directly incorporate our previous estimate of $\boldsymbol{u}$:

$$(\widehat{\boldsymbol{M}}^{(k)}, \widehat{\boldsymbol{d}}^{(k)}, \widehat{\boldsymbol{\zeta}}^{(k)}) = \underset{\boldsymbol{M}, \boldsymbol{d}, \boldsymbol{\zeta}}{\arg\min} \ \ell(\boldsymbol{d}) + \gamma_1^{(k)} \|\boldsymbol{\zeta}\|_1 + \gamma_2^{(k)} \|\boldsymbol{M}\|_F^2 + \gamma_3^{(k)} \|\boldsymbol{d}\|_2^2 \tag{12}$$

$$\text{s.t.} \quad -\boldsymbol{\zeta} \le \text{diag}(\boldsymbol{S}\boldsymbol{M}\boldsymbol{R}^T) - \boldsymbol{Q}_\Gamma \boldsymbol{d} - 2\boldsymbol{R}\boldsymbol{M}\widehat{\boldsymbol{u}}^{(k-1)} \le \boldsymbol{\zeta}$$

$$\boldsymbol{\zeta} \ge \boldsymbol{0}, \quad \boldsymbol{M} \succeq \boldsymbol{0}.$$

We can then update our estimate of $\boldsymbol{u}$ as before:

$$\widehat{\boldsymbol{u}}^{(k)} = \frac{1}{2}(\widehat{\boldsymbol{M}}^{(k)} \boldsymbol{R}^T \boldsymbol{R}\widehat{\boldsymbol{M}}^{(k)} + \alpha^{(k)} \boldsymbol{I})^{-1} \widehat{\boldsymbol{M}}^{(k)} \boldsymbol{R}^T (\boldsymbol{a}_{\widehat{\boldsymbol{M}}^{(k)}} - \boldsymbol{Q}_\Gamma \widehat{\boldsymbol{d}}^{(k)}).$$

Note that we allow the regularization parameters to change across iterations. In practice we fix $\gamma_1^{(k)}$, $\gamma_2^{(k)}, \gamma_3^{(k)}$, and $\alpha^{(k)}$ for all iterations $k \ge 1$, but we do consider an alternative set of parameters for the initialization ($k = 0$). This is somewhat natural since the initialization step actually involves solving a slightly different optimization problem.

### 3.6 Identifiability of the metric and ideal point

We conclude our description of our approach with a brief discussion of the degree to which the ideal point $\boldsymbol{u}$ and metric $\boldsymbol{M}$ are potentially identifiable.

**Proposition 1.** *For a fixed $\boldsymbol{M} \in \mathbb{R}^{D \times D}$, the ideal point $\boldsymbol{u}$ is identifiable if and only if $\boldsymbol{M}$ is (strictly) positive definite.*

The proof is provided in the supplementary material and is similar to the proof of Proposition 3 in [37]. This result is not surprising, as if $\boldsymbol{M}$ is rank-deficient, any part of $\boldsymbol{u} \in \ker(\boldsymbol{M})$ will be not recoverable. We note that, since we desire our constraint set to be closed, our estimation procedure enforces a positive *semi*-definite constraint. In practice, if $\boldsymbol{M}$ is ill-conditioned, we may estimate a solution which is rank-deficient (which ignores the eigenvectors corresponding to relatively small eigenvalues), and thus the portion of $\boldsymbol{u}$ in the span of these eigenvectors may be extremely difficult to

estimate. Note, however, that due to the influence of $M$, the unidentifiable portion of $u$ also plays little role in determining the underlying preferences. In recognition of this, we typically quantify our estimation performance in terms of $\|\widehat{u} - u\|_M$.

We also note that, at least in the noise-free setting considered in this paper, even if the metric is fully identifiable, it will only be so up to a constant scaling factor. However, note that a constant scaling factor does not change the learned ideal point or ranking of items. To see why this is true, note that for any $M$ and $d$ satisfying the constraint in (4), rescaling $M$ and $d$ (and hence $\delta_\Gamma$) by an arbitrary constant $c > 0$ will yield another valid solution. However, it is relatively easy to show that for arbitrary scaling of $M$ and $\delta_\Gamma$, the estimate provided by (5) of $\widehat{u}$, as well as the resulting ranking of the items, remains unchanged.

Finally, we also note that when recovering $M$, if a subset of the eigenvalues of $M$ are equal or relatively close, it becomes impossible, or at least more difficult, to distinguish among the specific eigenvectors. In this case, our estimate may swap the order of the eigenvectors or learn different eigenvectors that span a similar space to the original, but can be quite different. As with scaling, this has little impact on estimating $u$ or in terms of the resulting rankings, but plays an important factor in determining the appropriate evaluation metrics.

## 4 Experiments

### 4.1 Synthetic experiments

In this section, we demonstrate the effectiveness of the joint estimation on synthetically generated data[2]. We assume *a priori* knowledge of an existing embedding of items and estimate $u$ and $M$. In each simulation, $N$ items $x_1, \ldots, x_N$ are generated uniformly on the hypercube $[-2, 2]^D$ and one user $u$ is generated uniformly on $[-1, 1]^D$. A positive definite matrix $M$ is generated by $M = L^T L$, where the entries of $L \in \mathbb{R}^{D \times D}$ are drawn from the standard normal distribution. Noiseless comparisons are chosen uniformly without repetition and used for joint estimation.

Certain conditions are imposed on the matrix $M$: 1) The Frobenius norm of $M$ exceeds a small chosen threshold $\epsilon_F$, 2) The smallest singular value of $M$ is larger than a small chosen threshold $\epsilon_S$, and 3) The fraction $\|Mu\|_2/\|u\|_2$ exceeds a small chosen threshold threshold $\epsilon_P$. $\epsilon_F$ and $\epsilon_S$ are imposed to guard against numerical instabilities while $\epsilon_P$ is necessary to ensure that $u$ is identifiable. For all synthetic experiments, the chosen values were $\epsilon_F = 0.5$, $\epsilon_S = 0.25$, and $\epsilon_P = 0.2$.

We define the user's ideal point reconstruction error (UR error) as $\|\widehat{u} - u\|_M^2 / \|u\|_M^2$. Letting the eigendecompositions of $M$ and $\widehat{M}$ be $V \Lambda V^T$ and $\widehat{V}\widehat{\Lambda}\widehat{V}^T$, respectively, we define the weighted eigenstructure reconstruction error (WER error) as $\|\Lambda \odot |V^T \widehat{V}| - \Lambda\|_F^2 / \|\Lambda\|_F^2$, where $\odot$ denotes element-wise multiplication and $|A|$ takes the element-wise absolute value of $A$. When $\widehat{M}$ is recovered to be a scaled version of $M$, we expect the diagonal elements of $|V^T \widehat{V}|$ to be 1. In all cases when the WER error is small, $M$ is recovered well. However, there are instances in which a high value of the WER error does not imply a poor estimate of $M$. For example, (large) repeated eigenvalues in $M$ would result in a large WER error if the eigenvectors in $\widehat{V}$ differed, but spanned the same space. Our synthetic data avoids this, but care is needed to quantify performance in general.

**Single-step estimation** In the first simulation, we show the improvement in estimation as the number of comparisons increases. For a fixed number of comparisons, we perform 100 trials and report the median UR error and WER error, and interpolated median of the fraction of the top 10 closest items to $u$ identified for $D = 2, 5$, and 10. Since the fraction of the top 10 items is discrete, we utilize the interpolated median in place of the usual median. In all cases, we include the 25% and 75% quantiles. For each trial, we generate a new metric, ideal point, and $N = 100$ items.

As shown in Fig. 1, when a small number of comparisons are used for joint estimation, the UR and WER error are large, while the fraction of top 10 items correctly identified is small. As the number of comparisons increases from 10 to 500, the UR and WER errors decrease rapidly, while the fraction of top 10 items increases rapidly.

In the second simulation, we compare the performance of our algorithm against two algorithms that assume Euclidean distance to estimate the ideal point. **Euclidean Algorithm 1** is an adaptation of

---

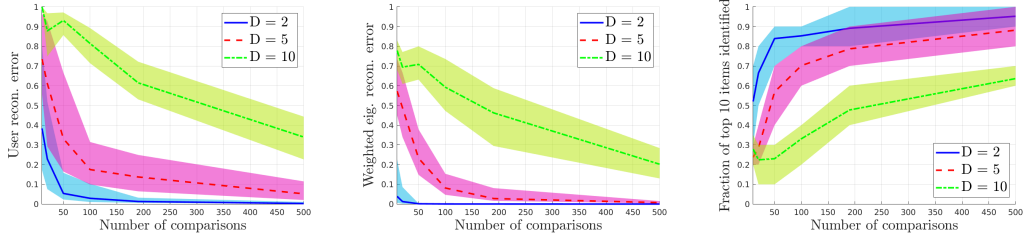

Figure 1: Median UR error, WER error, and interpolated median fraction of top 10 items identified over 100 trials, plotted with 25% and 75% quantiles. As the number of comparisons grows, both UR and WER error decrease to 0 as the fraction of top 10 items increases to 1 for all $D$. Regularization parameters: $\gamma_1 = 2, \gamma_2 = 0.002, \gamma_3 = 0.001, \alpha = 1$.

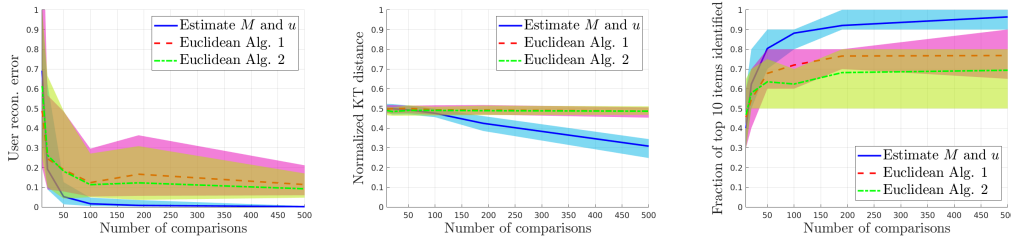

Figure 2: Comparison of singe-step estimation against Euclidean Algorithms 1 and 2 when the true distance metric is $\boldsymbol{M} \neq \boldsymbol{I}$. Regularization parameters: $\gamma_1 = 2, \gamma_2 = 0.002, \gamma_3 = 0.001, \alpha = 1$.

our single-step algorithm to solve for only the distances $\boldsymbol{d}_e$ between items and the ideal point:

$$(\widehat{\boldsymbol{d}}_e, \widehat{\boldsymbol{\zeta}}) = \arg\min_{\boldsymbol{d}, \boldsymbol{\zeta}} \ell(\boldsymbol{d}_e) + \gamma_1 \|\boldsymbol{\zeta}\|_1 + \gamma_2 \|\boldsymbol{d}_e\|_2^2 \tag{13}$$
$$\text{s.t.} \quad -\boldsymbol{\zeta} \leq (\boldsymbol{I} - \boldsymbol{R}\boldsymbol{R}^\dagger)(\text{diag}(\boldsymbol{S}\boldsymbol{R}^T) - \boldsymbol{Q}_\Gamma \boldsymbol{d}_e) \leq \boldsymbol{\zeta}, \qquad \boldsymbol{\zeta} \geq \boldsymbol{0}.$$

From here, we can solve for $\widehat{\boldsymbol{u}}$ by replacing replacing $\widehat{\boldsymbol{M}}$ with $\boldsymbol{I}$ in (9). **Euclidean Algorithm 2** is the approach in [21], which directly solves a convex program for $\boldsymbol{u}$ from the paired comparisons.

We sweep the performance for all three algorithms for $D = 2$ over different numbers of comparisons between 10 and 500. For a fixed number of comparisons, we perform 100 trials and report the median (or interpolated median) and 25% and 75% quantile for UR error, normalized Kendall's Tau distance, and the fraction of top 10 items identified. For each trial, we generate a new metric and ideal point and $N = 100$ new items. As seen in Fig. 2, our algorithm outperforms both algorithms that assume a Euclidean distance metric by recovering a more accurate ideal point, ranking of items, and fraction of top $K$ items. The same experiment was performed when $\boldsymbol{M} = \boldsymbol{I}$ for all trials with very little loss in performance by using our algorithm (see the supplementary material for further details).

**Alternating estimation** We now explore the potential improvements that can be attained by our alternating estimation procedure. For $D = 5$, we fix an ideal point, metric, and a set of $N = 100$ items, and vary the number of comparisons. For a fixed number of comparisons $P$, we run 100 trials, where we select $P$ new comparisons at random. We then run the alternating descent until the difference in the user reconstruction error between successive iterations is less than $10^{-3}$, with a maximum number of iterations set to 100. We report the median and 25% and 75% quantiles for the initial and final UR error in Fig. 3. We observe that alternating estimation does not improve the estimate of $\boldsymbol{u}$ much when the number of comparisons is small ($< 40$) or large

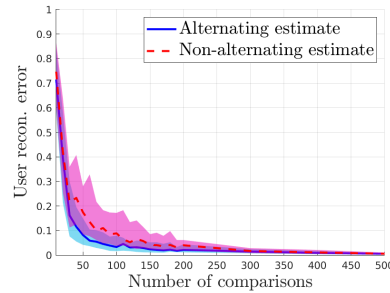

Figure 3: UR error for single-step and alternating estimation. Regularization parameters: $\gamma_1^{(0)} = 2, \gamma_2^{(0)} = 0.002, \gamma_3^{(0)} = 0.0001, \alpha^{(0)} = 1; \gamma_1^{(k)} = \frac{2}{3}, \gamma_2^{(k)} = \frac{1}{15}, \gamma_3^{(k)} = \frac{7}{1500}, \alpha^{(k)} = \frac{1}{2}$ for $k \geq 1$.

$(> 200)$. In the first regime, the comparisons do not reveal enough information to reliably recover $\boldsymbol{u}$, while in the second regime, the number of comparisons is sufficient to make the single-step estimation very accurate. The alternating method offers steady improvement in the intermediate regime, and is able to successfully reduce the error nearly 60%.

## 4.2 Graduate admissions dataset

We now apply our models to two PhD program admissions datasets from Georgia Tech School of Electrical and Computer Engineering. The *Unranked Candidates* dataset consists of over 3,000 applicants in three categories: 1) admitted with fellowship, 2) admitted, and 3) denied admission. The applicants are not ranked, so the only paired comparisons we can form are across categories. We assume that fellowship recipients are preferred to admitted candidates, who are preferred to denied candidates, so for $N_F$ fellowship recipients, $N_A$ admitted candidates, and $N_D$ denied candidates, we can form at most $N_F(N_A + N_D) + N_A N_D$ comparisons. For each applicant, we have access to five features: GPA, GRE quantitative, verbal, and analytical writing scores, and a letter of recommendation (LoR) score. Each candidate's GPA is normalized to a 4.0 scale. The GRE verbal and quantitative scores are integers between 130 and 170, inclusive, while the GRE writing score is from 0 to 6 in 0.5 increments. Each candidate submitted at most three letters of recommendation, each of which is scored on a scale of 0 to 3. The scores are averaged and then exponentiated to obtain a LoR score between 1 and $e^3 \approx 20.09$.

The *Ranked Candidates* dataset consists of 88 applicants who are scored on a scale of 1 to 10, with 1 being the most preferred and 10 being least preferred. The top 11 candidates are uniquely rank ordered, and the rest of the candidates are sorted into 8 bins of various sizes. We only form comparisons between candidates with different scores, so two candidates with the same score are not compared. For each applicant, we have access to the same features except for letter of recommendation scores.

***Unranked Candidates*** We begin by noting that the features being used in this model are inherently restrictive. Applicants are evaluated on many criteria beyond the features included, which can lead to occasional unexpected results. For instance, there exist large subsets of denied candidates whose average GRE scores are higher than those of a some fellowship recipients, which might indicate that a lower GRE score is more favorable, occasionally leading to rather unusual ideal point placement. Furthermore, the learned ideal point may fall outside of appropriate ranges (e.g., GPA > 4), which indicates an extremely strong preference for that feature. However, one can interpret values outside of appropriate ranges as a preference for the maximum/minimum appropriate score. In reality, we would expect that the optimal set of features should be the maximum value for all possible features. Furthermore, of the five features, we suspect that the GRE verbal score should likely matter the least, followed by the GRE quantitative score, as applicants from across the categories score similarly on these two GRE sections. Our expectation is that the most significant features should be some combination of GRE writing, GPA, and LoR. With this in mind, we use our algorithm to learn relevant feature interactions and confirm our hypothesized ordering of the importance of features via the learned metric. We take $N_F = 33$, $N_A = 33$, and $N_D = 34$, form all 3333 possible comparisons, and learn the ideal point $\boldsymbol{u}$ and metric $\boldsymbol{M}$ using a subset of all features.

When all five features are used to learn $\widehat{\boldsymbol{u}}$ and $\widehat{\boldsymbol{M}}$, our hypothesized ordering of importance for the features is correct. The three most significant features are a weighted difference between GPA and GRE writing, a weighted sum of GPA and GRE writing, and the LoR score. The learned ideal scores are 158.08 GRE verbal, 162.50 GRE quantitative, 4.68 GRE writing, 4.06 GPA, 15.28 LoR score. As seen in Fig. 4, when the GRE verbal and quantitative scores features are used, the learned feature interactions are a weighted difference (eigenvector 1) and sum (eigenvector 2) of GRE verbal and quantitative scores. The structure of the learned metric seems to make intuitive sense, indicating that in order to compensate for a slightly lower GRE quantitative score, one must score significantly higher on the GRE verbal section.

***Ranked Candidates*** Since ranking information is partially available in this dataset, we record the fraction of top $K = 11, 17$, and 22 candidates correctly identified as the number of comparisons increases using all four features. For a fixed number of comparisons, we perform 20 trials and report the mean and standard deviation of the fraction of the top $K$ candidates correctly identified in Fig. 5. The fraction of the top $K$ candidates correctly identified for $K = 11, 17$, and 22 increases rapidly as the number of comparisons increases. With less than 20% of the total number of comparisons, we can identify over 90% of the top 22 and 17 candidates and over 80% of the top 11 candidates correctly.

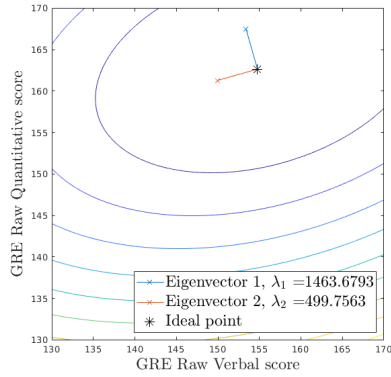

Figure 4: Level sets for learned metric for *Unranked Candidates* GRE verbal and quantitative scores. Regularization parameters: $\gamma_1 = \frac{1}{650}, \gamma_2 = \frac{1}{6500}, \gamma_3 = \frac{2}{65} \cdot 10^{-6}, \alpha = 1$.

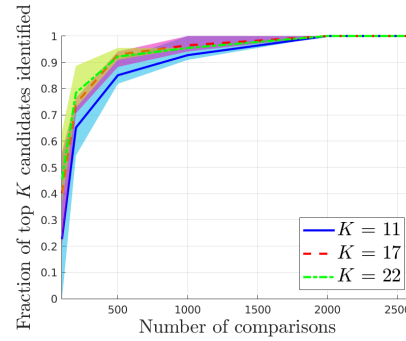

Figure 5: Fraction of top 11, 17, and 22 of ranked candidates identified. Regularization parameters: $\gamma_1 = \frac{3}{800}, \gamma_2 = \frac{1}{8000}, \gamma_3 = \frac{5}{8} \cdot 10^{-11}, \alpha = 1$.

## 5 Discussion

In this paper, we develop a method for jointly learning a user's ideal point and an underlying distance metric from paired comparisons. The metric captures feature interactions and their relative significance to users, neither of which are captured by the traditional Euclidean metric. We demonstrate our algorithm can correctly identify the ideal point and metric and can correctly rank graduate admission candidates and determine feature interactions on real-world data. We conclude by noting that in the Euclidean setting, adaptive querying schemes have been shown to enable dramatic reductions in the required number of comparisons [9, 42]. We expect similar gains are possible in our setting. Developing novel methods for adaptively selecting comparisons to maximize the amount of information collected about both $u$ as well as $M$ is an important avenue for future research.

## Broader Impact

In an increasingly diverse set of contexts, automated ranking systems are playing an increasing role in society. We naturally have an interest in ensuring that these systems are as accurate as possible and not undermined by poor modeling assumptions. If used to augment such systems, our research could be used to develop improved models for how users implicitly process the features of the ranked items by learning a metric. In particular, in addition to providing a more powerful and flexible model of preference, the metric illuminates how features are combined and the ordering of importance of the combined features, providing both additional insight as well as helping to better identify user desires and rankings.

In certain applications, such as in hiring or admissions committees, there is also the potential to apply our techniques to discover if the metric of an evaluator is significantly influenced by certain factors (e.g., race, gender, sexual orientation, religion, etc.) which we do not wish to influence our decisions. This information may be useful as feedback to the evaluator or alternatively, to explicitly compensate for the existence of such influence (e.g., by re-ranking candidates using a modified metric that eliminates dependence on those factors).

However, this optimistic assessment of the potential of our approach should be tempered by two important caveats. First, in the absence of any strong theoretical guarantees, we are unable to provide any notion of confidence intervals or statistical significance in the learned parameters, and so one must evaluate the learned models with a degree of caution and awareness that the learned parameters may be the result of chance variation in a small dataset. More generally, there is no notion of causal influence in our model, and it is always possible that the observed dependence on the features is the result of correlations with some unobserved latent factors. Second, due to both the potential for inaccuracies in the learned parameters as well as the personal nature of the revealed preferences, as with any system involving personal data, privacy remains a significant concern.

## Acknowledgments and Disclosure of Funding

This work was supported, in part, by the Alfred P. Sloan Foundation.

## Footnotes

[1]Note that this follows from the general identity $\text{vec}(\boldsymbol{A}\boldsymbol{B}\boldsymbol{C}) = (\boldsymbol{C}^T \otimes \boldsymbol{A})\text{vec}(\boldsymbol{B})$.

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
