[Supplementary Material]

# Supplementary Material

## A   Proof of Proposition 1

**Proposition 1:** For a fixed $M \in \mathbb{R}^{D \times D}$, the ideal point $u$ is identifiable if and only if $M$ is (strictly) positive definite.

**Proof.** Let $w \in \mathbb{R}^D$ be arbitrary. Note that for any point $x \in \mathbb{R}^D$, one can easily show that

$$\|x - u\|_M^2 = \|x - w\|_M^2 \tag{14}$$

if and only if

$$\langle 2x - u - w, M(u - w) \rangle = 0. \tag{15}$$

This follows simply by expanding the expressions on both sides of (14) and rearranging the terms to obtain (15).

We now show that if $u$ is identifiable then $M$ is strictly positive definite. Suppose for the sake of a contradiction that $M$ is not strictly positive definite, i.e., that there exists a non-zero $v \in \mathbb{R}^D$ such that $Mv = 0$. Let $w = u - v$. Then, by (15)

$$\langle 2x - u - w, M(u - (u - v)) = \langle 2x - u - w, Mv \rangle = 0.$$

From this we can show that, $\|x - u\|_M^2 = \|x - (u - v)\|_M^2$. This is a contradiction since $u$ cannot be identifiable as $w = u - v \neq u$ would yield identical observations.

We now show that if $M$ is positive definite then $u$ is identifiable. Suppose that $w \in \mathbb{R}^D$ satisfies $\|x - u\|_M^2 = \|x - w\|_M^2$ for all $x \in \mathbb{R}^D$. From (15) we have that because $\langle 2x - u - w, M(u - w) \rangle = 0 \ \forall x \in \mathbb{R}^D$, it must be the case that $M(u - w) = 0$. If $M$ is positive definite, then it must be the case that $u - w = 0$, and hence $w = u$. $\square$

## B   Additional Synthetic Simulation Results

**Additional results for single-step estimation**   For the single-step estimation experiment found in Section 4.1, we also quantify algorithm performance via the normalized Kendall's Tau distance and the fraction of top 5 and 20 items correctly identified. The median (or interpolated median) and 25% and 75% quantiles are reported in Fig. 6. While the normalized Kendall's Tau distance decreases for $D = 2, 5$, and $10$, it does so rather slowly. This is due to the fact that many items are very similar to each other in terms of their distance from $u$, and hence getting the exact ordering of *all* items correct is rather difficult. However, the performance in identifying the top $5, 10$, and $20$ items is strong, which indicates that the algorithm is in fact learning which items are important.

Figure 6: Median normalized Kendall's Tau distance and interpolated median fraction of top 5 and 20 items identified over 100 trials, plotted with 25% and 75% quantiles. Regularization parameters: $\gamma_1 = 2, \gamma_2 = 0.002, \gamma_3 = 0.001, \alpha = 1$.

**Single-step estimation when $M = I$**   We demonstrate the effectiveness of our algorithm when $M = I$ and compare performance with **Euclidean Algorithm 1** and **Euclidean Algorithm 2** as defined in Section 4.1. We sweep the performance for all three algorithms for $D = 2$ over different numbers of comparisons between $10$ and $500$. For a fixed number of comparisons, we perform $100$ trials and report the median (or interpolated median) and 25% and 75% quantile for UR error, normalized Kendall's Tau distance, and the fraction of top 5, 10, and 20 items identified. For each trial, we generate a new metric and ideal point and $N = 100$ new items. As seen in Fig. 7, there is no significant loss in performance when using our algorithm, especially as the number of comparisons increases. Thus, adding the additional flexibility to allow for $M \neq I$ does not seem to result in any significant penalties, even when $M$ is in fact $I$.

Figure 7: Comparison of singe-step estimation against Euclidean Algorithms 1 and 2 when the true distance metric is $\boldsymbol{I}$. Regularization parameters: $\gamma_1 = 2, \gamma_2 = 0.002, \gamma_3 = 0.001, \alpha = 1$.

Figure 8: Median WER error, normalized Kendall's Tau distance, and interpolated median for top $5, 10$, and $20$ items for single-step and alternating estimation. Regularization parameters: $\gamma_1^{(0)} = 2, \gamma_2^{(0)} = 0.002, \gamma_3^{(0)} = 0.0001, \alpha^{(0)} = 1; \gamma_1^{(k)} = \frac{2}{3}, \gamma_2^{(k)} = \frac{1}{15}, \gamma_3^{(k)} = \frac{7}{1500}, \alpha^{(k)} = \frac{1}{2}$ for $k \geq 1$.

**Additional results for alternating estimate** For the alternating estimation experiment found in Section 4.1, we also quantify algorithm performance via the WER error, normalized Kendall's Tau distance, and fraction of top $5, 10$ and $20$ items correctly identified. The median (or interpolated median) and $25\%$ and $75\%$ quantiles are reported in Fig. 8. In the intermediate regime (between $40$ and $200$ comparisons), the alternating estimate generally improves the WER error and fraction of top $K$ items identified. The normalized Kendall's Tau distance remains relatively the same for all comparisons, but the improvement in the fraction of top $K$ items indicates that the algorithm improves in identifying the which items are close to the ideal point.

## C   Data Pre-processing

*Unranked Candidates* **dataset pre-processing** The *Unranked Candidates* dataset is originally comprised of $3,789$ total applicants, with $191$ admitted with fellowship, $530$ admitted without

Table 1: Feature interactions and corresponding eigenvalues for the *Unranked Candidates* dataset for $N_F = 33, N_A = 33, N_D = 34$ and 3333 comparisons. Regularization parameters: $\gamma_1 = \frac{1}{650}, \gamma_2 = \frac{1}{6500}, \gamma_3 = \frac{2}{65} \cdot 10^{-6}, \alpha = 1$.

| Feature interactions in $\widehat{M}$. | |
| --- | --- |
| $\lambda_1 = 1991$ | $0.909$ GRE writing $-\ 0.392$ GPA |
| $\lambda_2 = 1971$ | $0.919$ GPA $+\ 0.393$ GRE writing |
| $\lambda_3 = 1178$ | $0.982$ LoR |
| $\lambda_4 = 861$ | $0.942$ GRE quant $-\ 0.310$ GRE verbal |
| $\lambda_5 = 286$ | $0.942$ GRE verbal $+\ 0.319$ GRE quant |

fellowship, and 3068 denied candidates. Ten raw features are associated with each candidate (Self-reported GRE analytical writing, self-reported GRE verbal, self-reported GRE quantitative, official GRE analytical writing, official GRE verbal, official GRE quantitative, GPA, and up to three scored letters of recommendation). Some candidates have missing entries for some of the ten raw features. Depending on which features are used to generate input data for the algorithm, we remove candidates with relevant missing data. If GRE scores are used, for each candidate, we take the official GRE scores to be the true GRE scores. If the official GRE scores are missing, then we take the self-reported scores. The raw GPA scores are already normalized on a 0 to 4 scale, but the normalization resulted in some unusable entries. If the GPA feature is used, we only keep candidates with GPAs between 1 and 4. The LoR score is computed as described in Section 4.2. In all, there are 3305 candidates with no missing entries (176 admitted with fellowship, 455 admitted candidates, and 2674 denied candidates).

***Ranked Candidates* dataset pre-processing**  The *Ranked Candidates* dataset originally contains 89 candidates with four raw features (GRE analytical writing, GRE verbal, GRE quantitative, and GPA). For this dataset, there is only one GRE score available to us, so there is pre-processing needed to discern between self-reported and offiical. There is one candidate with missing raw features who is discarded, leaving us with 88 usable candidates.

## D   Additional Experimental Results

**Additional results for *Unranked Candidates* dataset**  As reported in Section 4.2, the ideal point and metric is learned using a set of 100 candidates ($N_F = 33$, $N_A = 33$, and $N_D = 34$) and all possible comparisons (3333). The significant feature interactions are reported in Table 1, along with the corresponding eigenvalues. The weighted difference and sum of GPA and GRE writing score are the top two feature interactions and are almost equally important, followed by the LoR score and the weighted difference between GRE quantitative and verbal scores. The most insignificant feature interaction is the weighted sum of the quantitative and verbal scores.

Using the same number of candidates and comparisons, we also learn feature interactions and ideal points for pairs of features. For all pairs of features aside from GRE verbal vs. GRE quantitative (presented in Section 4.2), we display the level sets for the learned metric in Fig. 9. We again note that learning the ideal point with inherently restrictive features leads to unexpected behavior. In many cases, the ideal point value falls well outside of the allowed range for many of the features. For example in the GRE quantitative vs. GPA pair, the ideal GPA is  35, which is much larger than 4. In these cases, the fact that the ideal value is higher than the maximum allowed values indicates that the larger the score, the better. This is consistent with our expectation that the optimal set of features should be the maximum value for all possible features. Many pairs of features do not have meaningful learned interactions, but pairs of features such as GRE writing vs. GPA do have some meaningful interaction.

**Additional results for *Ranked Candidates* dataset**  For the *Ranked Candidates* dataset, we also record the the normalized Kendall's Tau distance for the top 11 candidates. We choose to evaluate the ranking of the top 11 candidates because these candidates are the ones most likely to be admitted. The median normalized Kendall's Tau distance and 25% and 75% quantiles can be found in 10. As the number of comparisons increases, we are able to extremely accurately predict the exact ranking of the top 11 candidates.

Figure 9: Level sets for pairs of features for *Unranked Candidates* dataset.

Figure 10: Normalized Kendall's Tau distance for top 11 ranked candidates identified. Regularization parameters: $\gamma_1 = \frac{7}{6002}, \gamma_2 = \frac{1}{6002}, \gamma_3 = \frac{2}{6002} \cdot 10^{-4}, \alpha = 1$.

The learned metric using all 2610 comparisons does not exhibit any meaningful feature interactions. GPA and GRE writing are the top two features with roughly equal eigenvalues, followed by GRE quantitative. The GRE verbal score is the least significant feature. This is consistent with our expected order of significance of features for candidates.

## E  Optimization parameter tuning

The reported parameters for the synthetic and real-world data experiments were obtained through manual tuning. We did not perform an exhaustive search over parameter combinations for any of the synthetic or real-world data experiments. The reported parameters appear to be very specific, but this is due to a change in notation, and hence a re-normalization of parameters. We have found that the algorithm is relatively robust to parameter values and defer developing a method for picking optimal or near optimal parameters to future work.