[Reviews · NeurIPS 2020]

Review 1

Summary and Contributions: This paper considers the recommendation problem, where a goal is represented by an ideal point in some multi-dimensional space, and items are ranked based on their distance from this point. Rather than using a standard Euclidean distance, this paper proposes learning a Mahalanobis distance while estimating the ideal point based on pairwise preferences.

Strengths: - This work presents a very elegant approach for an extremely popular problem with high novelty and broad relevance. - The presentation is very clear and easy to follow. - By relying on pairwise comparisons, this model also depends on easier to obtain data. - The algorithm is evaluated on synthetic and real dataset. - The motivation and broader impact is thoughtfully discussed

Weaknesses: - The application of recommendation to candidate admissions is, while very important as an example of fairness in AI, a little uncompelling from a recommendation evaluation perspective. The paper would be stronger if the method were also tested on a classic recommendation problem. This also means that future comparisons with this method as a baseline will be challenging (it will require re-implementation of the method as the authors have not indicated a willingness to share their code. While addressed in the author feedback, it is still a comparison I would like to see. - The authors note a lack of theoretical guarantees, which is a little disappointing especially given the iterative EM nature of the solution.

Correctness: This reviewer did not check the algorithmic derivations in detail. However, the experimental validation is appropriate and correct. The claims of how the method works are well supported.

Clarity: The paper is very clearly written, thoughtfully introducing concepts in a way that does not feel rushed. This is an exceptional presentation for a method that needs to introduce many concepts and steps within the tight limits of NeurIPS.

Relation to Prior Work: Prior work is thoroughly and clearly discussed, with contrasts drawn as required.

Reproducibility: Yes

Additional Feedback: In terms of reproducibility, the real world dataset may or may not be available publicly, we do not know. This may limit the possibility to reproduce the specific results, or for future work to compare directly to this approach. How were the regularization parameters set? And, trivially, Figures 1 and 5 could be shown on a logarithmic scale.


Review 2

Summary and Contributions: This paper proposes two algorithms that use pairwise comparisons of items to learn both user preference vectors and a Mahalanobis distance over the space of preference vectors (which are in the same space as the items) under the ideal point model. The paper presents experimental results on synthetic data as well as graduate admissions data, showing that the proposed methods can rank graduate school candidates and find interesting feature interactions.

Strengths: This paper is very well-written and the proposed algorithms are very cleanly explained. Researchers working on pairwise comparisons and preference learning should find this paper to be interesting and valuable.

Weaknesses: Update: The authors provided thoughtful feedback on noise considerations that addresses the main weakness mentioned below from my original review. Provided that this discussion on the noise model is incorporated into the final text, I think the paper will be substantially stronger. Additionally, the authors provide detailed feedback for other weaknesses that other reviewers brought up. Altogether, the additional discussion makes the paper much more thorough. In light of this, I am increasing my score of the paper from 7 to 8. One unresolved issue though: one of the reviewers pointed out that the ideal GPA of 4.06 is greater than 4 despite the GPA being normalized to a 4.0 scale. It would be helpful to clarify why this happens (e.g., I know some universities have a 4.0 scale but allow A+'s to be greater than 4.0, etc), and what exactly you mean by normalizing (e.g., dividing by the max which would undo the A+'s being greater than 4.0 for some schools, or changing some other school's scaling to be up to 4.0 in the case of, say, MIT which has a scale out of 5). ----- The main weakness I see in this paper is that the algorithms are motivated in a slightly heuristic fashion based on a noiseless model, and then the authors allow the solution form to be slightly different to tolerate deviations from the noiseless model. This leaves the question of whether there's a cleaner way to model the Bernoulli noise (simplest case would be where the noise is the i.i.d. across all comparisons) where learning preferences and the distance just amount to maximum likelihood? If so, in what way does this resemble the algorithms proposed?

Correctness: Yes, the claims and experiments look correct.

Clarity: Yes, the paper is very well-written and the exposition is quite clean.

Relation to Prior Work: Yes, the authors relate their work to existing literature.

Reproducibility: Yes

Additional Feedback: After reading the author feedback/other reviews: please incorporate the discussion points from the author feedback into the main paper. Also, see the GPA comment (originally from reviewer #3) and now also stated in the "weaknesses" question above.


Review 3

Summary and Contributions: The authors develop a method to jointly estimate ideal points and a Mahalanobis metric from noiseless pairwise comparisons and apply it to some synthetic and empirical datasets.

Strengths: The paper has a great treatment of the problem and analyzes the estimation problem at length. There are some interesting applications to synthetic and empirical data. The paper claims that the joint estimation problem of ideal points and Mahalanobis metric hasn't been solved before, which may be of independent interest.

Weaknesses: The over arching issue with the paper is a lack of prove theoretical claims or empirical wins over existing work, so this reads more like a deep dive on how to estimate a model along with some examples, which is interesting and possibly publishable in its own right, but doesn't stack up against contributions with empirical and theoretical wins. Other notes: -Focus on the noiseless case, which has less relevance in practice. -Lacks discussion on runtime (beyond pointing out that things are linear or convex) or how much data is needed for estimation -Seemed weird to me to include that the ideal GPA learned was 4.06, greater than the max GPA of 4.

Correctness: Everything is correct as far as I can tell.

Clarity: The paper is generally well written but I found some of the style in the related work to be dismissive of prior work. Claims like "The ideal point model is an intuitive and interpretable way to model preferences and has been empirically shown to exhibit superior performance compared to other models of preference" are overblown relative to the two citations that follow, you'd need some recent authoritative work on preference learning to justify a claim like that, especially when your paper doesn't include a comparison of performance against other methods.

Relation to Prior Work: The relation to prior work from the metric learning side is well discussed, but I think a lot of the existing work on estimating pairwise comparisons were swept over a bit lightly. The biggest need for prior work comparison is in the empirics section in my opinion, it's hard to know how useful this is when it's only compared against itself on a school choice dataset.

Reproducibility: Yes

Additional Feedback: I think this is good work but the crux of my score is on its relative theoretical and empirical strength to other work. Perhaps the estimation problem and formulation are of sufficient strength to justify publication from the metric learning perspective, which I have less expertise in and will defer to the other reviewers on if my review is an outlier. From the perspective of estimating models of comparison I think more work is needed to compare this method to existing ones and/or derive theoretical bounds on estimation and performance.


Review 4

Summary and Contributions: Authors propose method for simultaneously learning user preference vectors and an associated mahalanobis distance under the "ideal point" model of user choices and preference representation. They provide a convex program and two related algorithms for their learning problem. And they run experiments demonstrating the soundness of their algorithms on both synthetic data and datasets of graduate school applicants who have been accepted / rejected from academic programs.

Strengths: What is interesting, different, and perhaps surprising about this work is that given a set of item vectors { x_i } and a data set of user choices of the form "the user preferred item x_i over x_j", one can learn *both* the user's "ideal item" vector and her associated importance weighting of the item attributes (in the form of a mahalanobis distance) and combinations thereof, and as a convex program. Prior work has only focused on either learning the user's ideal point vector, or a mahalanobis distance, but not both simultaneously. The experiments demonstrate the authors' method works better than an alternative approach from the literature on small to modestly sized datasets when the underlying preferences are described by a mahalanobis distance that is not Euclidean.

Weaknesses: It should be pointed out that the ability to learn both a mahalanobis distance and user's ideal point is not totally novel since performing user/item matrix factorization from user pairwise preferences could be viewed as a case in which both the distance metric and the user representations are learned. But in matrix factorization, the mahalanobis distance gets pushed into the learned item embeddings rather than being explicitly captured in a separate parameter). Consequently, it is not clear that this method provides the community with truly new capabilities. And theoretically speaking, the existence of a convex program for this problem might have been expected, though it has not been published before. Second, by relying on semidefinite programming, the proposed method is not scalable to industrial recommender systems, at least "out of the box". But there are likely custom modifications that would make the proposed algorithm(s) useable in these settings. The authors might briefly discuss these since recommender systems are listed as a use case in the introduction.

Correctness: the claims appear correct

Clarity: yes, the paper is very well written and easy to follow

Relation to Prior Work: relationship to prior work is discussed and is clear.

Reproducibility: Yes

Additional Feedback: While the authors discuss identifiability, they might also want to discuss the sample complexity of learning ideal points and distances. How many comparisons are needed to estimate "u" and "M" well (for some definition of 'well'). Also, the authors consider learning "u" and "M" for a single user. But what if we have preference choices obtained from multiple users? If we assume all users have the same underlying "M", then we would need fewer choice observations per user to learn "M" well, would this in turn help us more quickly learn the ideal points for each of users?

[Author Response · NeurIPS 2020]

We thank the reviewers for thoughtful feedback. We are grateful that there is a consensus that we have developed (R1) "a very elegant approach for an extremely popular problem with high novelty and broad relevance" and that (R2) "researchers working on pairwise comparisons and preference learning should find this paper to be interesting and valuable." We also appreciate that the reviewers found the main result that it is possible to simultaneously learn an ideal point representation and a metric to be (R4) "interesting, different, and perhaps surprising." Below we respond to the most common critiques raised by the reviewers, indicating changes that we will make to the paper as appropriate. Furthermore, we note that we also plan to make our code available as soon as the review period concludes.

**Lack of theoretical guarantees (R1, R2, R3, R4).** We are in full agreement that strong theoretical guarantees are extremely important. While our work does not currently have such guarantees, we believe that our algorithm is significant in that it provides strong empirical evidence of the feasibility of simultaneous metric and ideal point recovery. As a starting point, we have explored borrowing ideas from [1], which gives theoretical error bounds on an alternating algorithm for a similar joint learning problem. However, due to the non-linear nature of our observation model, such an extension seems to be non-trivial at this point.

**Noise considerations (R2, R3).** In our derivation, we pose the problem in a noiseless environment only for simplicity. A similar derivation could be performed by assuming noisy distance estimates (although deriving the algorithm directly from the paired comparison model does not seem straightforward). Moreover, we emphasize that our algorithm can still work when differences in distances are corrupted by additive i.i.d noise. Such noise might arise from an imperfectly learned embedding or as a way of modelling response errors. The slack variables present in the optimization formulation not only increase the stability in the noiseless setting, but also allow for comparison outcomes to be violated due to such noise. Under common paired comparison noise models, we can also replace the loss function in (7) with the negative log-likelihood, provided that this is concave. For example, the loss function for logistic noise, which is equivalent to assuming comparison outcomes follow a Bradley-Terry model, is $\ell(\boldsymbol{d}) = \sum_{k=1}^{P} \log(1 + e^{-y_k (\boldsymbol{Q}_\Gamma \boldsymbol{d})_k})$ for $y_k = +1/-1$. While Bernoulli noise could be considered a simple noise model, we typically consider noise models where more the probability a comparison flips is inversely proportional to $|\boldsymbol{d}_i - \boldsymbol{d}_j|$, hence why we assume additive noise. We will include a more thorough discussion about noise considerations in our revision.

**Experiments: dataset choice, comparison with other techniques (R1, R3, R4).** We chose to test our algorithm on a graduate admissions dataset instead of traditional recommender system datasets (MovieLens, UT Zappos50K, etc). This was primarily because with traditional recommender datasets, we would have to construct an embedding first prior to preference/metric estimation. The source of error for unexpected outcomes would then be hard to pinpoint, as it could originate from model mismatch, a poorly learned embedding, or our estimation algorithm. Moreover, in such contexts we could not form interpretable hypotheses about which combinations of features should matter, further hindering our ability to evaluate the quality of the learned metric. The graduate admissions dataset alleviates these difficulties.
For similar reasons, we also did not compare our method against algorithms utilizing different models of preference. In our synthetic experiments, our priority was to quantify *estimation errors* in a setting where there was a known ground truth. While a more thorough comparison of our method with other models of preference on real-world data would certainly be interesting, explaining any gaps in performance (which might be due to either model-mismatch, estimation errors within the model, or a combination) would be difficult and would be beyond the scope of our paper.
Finally, R4 also points out that a similar process of implicitly learning a distance metric occurs in matrix factorization techniques. This is true and something we will emphasize more clearly in our revision, but we want to emphasize that this is different from our work in a key respect. Matrix factorization assumes user preference can be captured by (weighted) inner products, which is a *monotonic* function of the item features that assumes more (or less) of any particular feature is always better. This is distinct from our model, which allows for non-monotonic functions to describe user preference, meaning that there is a "sweet spot" for each item attribute.

**Practical implementation (R4).** As with any recommender system, practical considerations are important. R4 points out that solving a semidefinite program limits the capability of the algorithm to operate at an industry scale. We recognize this bottleneck and will add discussion on the necessity of practical tools, such as tailored SPD solvers or heuristically driven non-SPD approaches for large scale problems. Also mentioned was the idea of aggregating responses from different users and solving for their respective ideal points under the assumption that all users share the same metric. In our algorithm's current form, we cannot simply aggregate responses as each $\boldsymbol{d}$ vector is personalized for a specific user. However, an extension of our work would be incorporate multiple users and the constraints that arise from collecting (quantized) distance measurements between a set of ideal points and items in a fixed embedding. We consider this extension extremely interesting, but highly non-trivial, and defer it to future work.

Finally, we also plan to update the paper based on more detailed critiques, such as how regularization parameters were chosen (R1), or why the ideal point may slightly deviate outside of "acceptable" ranges (R3).

[1] Sheng Chen and Arindam Banerjee. An improved analysis of alternating minimization for structured multi-response regression. In *Advances in Neural Information Processing Systems*, pages 6616–6627, 2018.


[Meta-Review · NeurIPS 2020]

All of the reviewers are in agreement that this is a very nice paper which deals with a hot problem in an interesting fashion, has very interesting evaluations, and should be accepted to the conference.